# Fecal Microbiome Features Associated with Extended-Spectrum β-Lactamase-Producing Enterobacterales Carriage in Dairy Heifers

**DOI:** 10.3390/ani12141738

**Published:** 2022-07-06

**Authors:** Adar Cohen, Liat Poupko, Hillary A. Craddock, Yair Motro, Boris Khalfin, Amit Zelinger, Sharon Tirosh-Levy, Shlomo E. Blum, Amir Steinman, Jacob Moran-Gilad

**Affiliations:** 1Koret School of Veterinary Medicine, Hebrew University of Jerusalem, Rehovot 76273, Israel; adarkohen@gmail.com (A.C.); amit.shani@mail.huji.ac.il (A.Z.); sharontirosh@gmail.com (S.T.-L.); amirst@savion.huji.ac.il (A.S.); 2Microbiology, Advanced Genomics and Infection Control Application Laboratory (MAGICAL) Group, Department of Health Policy and Management, School of Public Health, Faculty of Health Sciences, Ben-Gurion University of the Negev, Beer-Sheva 8410501, Israel; liat.poupko@gmail.com (L.P.); motroy@post.bgu.ac.il (Y.M.); boriskh83@gmail.com (B.K.); giladko@post.bgu.ac.il (J.M.-G.); 3Division of Bacteriology and Mycology, Kimron Veterinary Institute, Bet Dagan 50200, Israel; shlomo.blum@mail.huji.ac.il

**Keywords:** antibiotic resistance, ESBL, cattle, microbiome, food safety, livestock health, One Health

## Abstract

**Simple Summary:**

Extended-spectrum β-lactamases (ESBLs) are a growing public health threat in terms of antimicrobial resistance (AMR), and one key human exposure point is through livestock and the food supply. Many factors can contribute to livestock carrying ESBLs, including feeding patterns, crowdedness, and the microbiome (bacterial ecosystem) of the animal. In this study, we observed that heifers from farms feeding calves with pooled colostrum had higher rates of ESBL carriage. Several genera of bacteria also differed between ESBL carriers and noncarriers. This study provides some potential directions for future research regarding ESBL carriage in heifers.

**Abstract:**

Extended-spectrum β-lactamases (ESBLs) are a growing public health threat, and one key human exposure point is through livestock and the food supply. Understanding microbiome factors associated with fecal ESBL carriage can help detect and ideally assist with controlling and preventing ESBL dissemination among livestock. The objective of this study was to investigate the diversity and composition of the heifer fecal microbiota in ESBL-producing Enterobacterales (ESBL-PE) carriers and noncarriers. A total of 59 fecal samples were collected from replacement heifers between 12 and 18 months old from eight dairy farms in central Israel. Genomic DNA was extracted, and 16S rRNA amplicon sequencing was performed (Illumina short reads), focusing on a comparison between 33 ESBL-PE carriers (55.9%) and 26 (44.1%) noncarriers. Samples were analyzed and compared using QIIME2 (DADA2 pipeline and taxonomic assignment with SILVA database) and associated R packages for alpha and beta diversity and taxonomic abundances. Alpha diversity (Shannon diversity) and beta diversity (unweighted UniFrac) showed no significant difference between ESBL-PE carriers and noncarriers. Heifers from farms feeding calves with pooled colostrum had higher ESBL-PE carriage rates than heifers from farms feeding with individual mother colostrum (*p* < 0.001). Taxonomical abundance analysis revealed that the most common bacterial phyla were Bacteroidetes (44%) and Firmicutes (38%). There was no significant difference in taxonomic composition between ESBL-PE carriers and noncarriers at the phylum and genus levels. However, LEfSe biomarker discovery analysis identified several genera which were significantly different between carriers and noncarriers. For example, Prevotellacaea, *Bacteroides*, Rikenellaceae, and uncultured Bacteroidales were more abundant in ESBL carriers than noncarriers. Some aspects of microbiota composition differ between ESBL carriers and noncarriers in dairy heifers, specifically the abundance of certain genera. Feeding with pooled colostrum may play a role in that assembly. These could potentially serve as markers of ESBL-PE carriage. However, further research is needed to determine whether these observed differences have a significant impact on colonization with ESBL-PE.

## 1. Introduction

Extended-spectrum β-lactamase (ESBLs) have become one of the most clinically and economically important antibiotic resistance mechanisms in human and veterinary medicine and are considered a serious threat to public health [1]. In search of the source of ESBL-producing Enterobacterales (ESBL-PE), production animals, including cattle, have raised concerns for being able to serve as a reservoir and transmission agent of ESBL-PE, due to their direct connection with the food chain [1,2,3]. Cattle are one of the main sources of animal protein for human consumption [4]. In the aspect of One Health, molecular typing investigations of different gut colonization samples suggested that cross-transmission among the farmers, livestock, and farm environment may be possible [2].

ESBL-PE have been isolated from cattle in North America, Europe, Asia, and the Middle East [3,5]. For example, in Switzerland, two independent studies showed 17.1% and 8.4% prevalence of ESBL-producing bacteria in healthy cattle, primarily *Escherichia coli* [3,6]. In Taiwan, 42.2% of *E. coli* isolates from beef carcasses produced ESBLs [7]. In Israel, a national survey of cattle in 2013 reported a prevalence of 23.7% ESBL-PE-positive cows, which was lowest among adult cows (age > 25 months) and highest among calves (age < 4 months) [5]. Another study in the United States (US) showed that 92% of beef calves become colonized by ESBL-PE at least once during their first year of life even without exposure to antibiotics [8].

The fecal microbiome has been extensively studied, and the wide variety of bacteria found in this microbiome is thought to play a chief role in maintaining the homeostasis of the gastrointestinal tract, as well as overall health [9]. Advances in technologies over recent years have generated a large amount of data on the composition and function of the rumen microbiota across a range of hosts and environments. Furthermore, there is increasing evidence that the lower gastrointestinal tract, which is primarily dominated by Firmicutes and Bacteroidetes, is also an important contributor to cattle health and production via maintenance of intestinal homeostasis, mucosal and lymphoid structure development, and activation of the host immunity [10,11]. Previously, a phenomenon termed “colonization resistance”, the prevention of colonization by exogenous bacteria including resistant potentially pathogenic microorganisms, was suggested [12]. For example, in mice, colonization with vancomycin-resistant enterococci (VRE) was inhibited by nutrient-depleting anaerobic bacteria in the colon, which limited the association of VRE with the mucus layer [13]. Furthermore, managing the fecal microbiota of cattle has implications for mitigating environmental impacts of pollution via manure management [10,14]. 

Despite the increasing recognition of the role of the intestinal microbiota in controlling the colonization of specific pathogenic and antibiotic-resistant bacteria, there is limited research investigating this role regarding colonization by ESBL-PE in both humans and animals. To the authors’ knowledge only one such study has been carried out in cattle. In this study, beef calves colonized by cefotaxime-resistant bacteria (CRB) had a fecal microbiome with higher abundance of *Fusobacteria*, *Elusimicrobia*, *Chlamydia*, and *Cyanobacteria* and lower abundance of spirochetes compared to calves not colonized by CRB [8]. Regarding human studies, microbiome differences were also found to be associated with ESBL-PE colonization. In a study conducted in French Guiana, the composition of the microbiota of ESBL-PE carriers was less diverse than in noncarriers [12]. In a more recent study from Thailand, where ESBL-PE colonization is high, the most notable difference was that the phylum Bacteroidetes was more abundant in ESBL-PE noncarriers [15]. In a study carried out in France, taxonomic and functional differences were also observed between the microbiome of ESBL-PE carriers versus noncarriers, including lower diversity in ESBL carriers than noncarriers [16]. Intestinal colonization is recognized as a key risk factor regarding ESBL-PE dissemination. Epidemiological and individual risk factors for intestinal colonization by ESBL-PE have been studied extensively; however, whether colonization is associated with significant changes in the composition of the rest of the microbiota is still understudied [12]. The objective of this study was to investigate the diversity and composition of the fecal microbiota in ESBL-PE carrier vs. noncarrier heifers in dairy farms in central Israel.

## 2. Materials and Methods

### 2.1. Demographic and Environmental Data Collection

The survey included eight dairy farms (18–20 heifers per farm) in the center of Israel (Figure 1). Farms ranged in size from 180 to 370 lactating cows, and all cows were from the Holstein Friesian breed. In each farm, cows were separated according to age. Lactating cows and heifers were typically separated by fences but under the same roof. Calves lived in a different area, usually at the edge of the farm. Data were collected by direct observation or by interviewing the farm managers. Included variables were stocking density (number of heads per square meter), cooling system, calf nutrition (pooled vs. individual colostrum), percentage of calf mortality and morbidity, percentage of udder morbidity (mastitis) in cows, the use of antimicrobial prophylaxis, bed type, use of summer and winter courts, geographical location, presence of other animals on the farm (including farm dogs), recent introduction of new calves to the farm, and veterinary care practices.

### 2.2. Fecal Sample Collection

Fresh fecal samples were collected through rectal picking from 157 randomly selected heifers (one sample per animal), aged 12–18 months. Collection took place after a routine health monitoring round conducted by the farm’s veterinarian. Immediately after collection, the collection medium was divided into two aliquots, with one aliquot stored at −80 °C for microbiome analysis, and the other aliquot freshly used for ESBL screening. Samples were collected between February and March 2020. The study was approved by the Hebrew University Ethics Committee (HU-NER-2020-016-A).

### 2.3. ESBL-Producing Enterobacterales Screening

Fresh fecal samples were first enriched in BHI broth and incubated at 37 °C for 16–18 h [17]. Samples were then plated onto CHROMagar ESBL plates (HyLabs, Rehovot, Israel), incubated at 37 °C for 16–18 h, and sub-cultured to obtain pure cultures. ESBL production was confirmed using the combination disc test according to the Clinical and Laboratory Standards Institute (CLSI) guidelines [18]. ESBL-producing isolates were then identified to the species level by matrix-assisted laser desorption/ionization time of flight (MALDI-TOF) mass spectrometry (MS) analysis on a Bruker Autoflex maX instrument (Bruker corporation, Berlin, Germany) as instructed by the manufacturer [19].

### 2.4. Samples for Microbiome Study

Fifty-nine samples were subjected to microbiome analysis. Our initial survey revealed 35% (55/157) ESBL-PE (range 0–100%, median 29%) colonization in all farms, of which 87.2% (48/55) were *E. coli.* In one farm, all sampled heifers were ESBL-PE- and *E. coli* ESBL (ESBL-E)-negative; in another farm, all sampled heifers were ESBL-PE- and ESBL-E-positive. In order to achieve similar numbers of cases and controls and equal presentation of farms, we picked between 6–9 heifers from each farm, half ESBL-PE-positive and half ESBL-PE-negative when possible. Through this sub-selection process, 33 ESBL-PE-positive and 26 ESBL-PE-negative samples were selected for microbiome analysis. 

### 2.5. DNA Extraction and 16S rRNA Gene Sequencing

DNA from swabs was extracted with the DNeasy PowerSoil kit without modifications (Qiagen, Hilden, Germany). The V4 region of 16S rRNA was amplified using 515F and 806R primers and processed on a MiSeq instrument per Illumina’s 16S protocol (using a Miseq V2-500 cycle kit to generate 2 × 250 paired-end reads) (Illumina, San Diego, CA, USA) [20].

### 2.6. Sequence Data Processing

Paired-end reads yielding a total of 3,513,274 bp were generated. The library size for the samples varied from 41,949 to 80,881 with a mean library size of 59,604 reads. After quality trimming wherein both forward and reverse reads (i.e., read pairs) were trimmed to 200 bp to remove adapters and low-quality read ends, as well as initial filtering, the library size varied from 27,728 to 54,982 reads, with a mean library size of 40,260 reads. All 59 samples were rarefied to 27,700 reads for downstream analyses. Adapter trimming was performed on Illumina’s built-in platform.

### 2.7. Taxonomic, Diversity, and Statistical Analysis

Raw sequenced amplicons were imported into the QIIME2 package (v2019.10) and analyzed by the DADA2 pipeline for quality control (QC), after removal of chimeric sequences. On average, 66.9% of reads were retained after QC (range 52.7–74.7%). Samples were then rarefied to 27,700 reads and subsequently assigned to taxonomic groups with the SILVA rRNA database (SILVA release 132, 99%, with the qiime2 classifier trained on the 515F/806R V4 region of 16S) [20].

Analyses performed included comparisons of relative abundance at the phylum and genus levels, comparisons of alpha and beta diversity, and differential abundance using linear discriminant analysis (LDA) effect size (LefSe) analysis. Regarding LefSe analysis, results were considered significant where the LDA score was greater than three [20]. For alpha diversity analysis, the Shannon diversity and Faith’s phylogenetic diversity (PD) indices were used. For beta diversity between populations, the unweighted UniFrac and Bray–Curtis metrics [21,22] were used. For beta-diversity statistical comparisons, PERMANOVA was used (999 permutations). For alpha diversity, statistical comparisons between two groups the Wilcoxon test was used, while, for statistical comparisons among more than two groups, the ANOVA test was used. Notably, data passed the Shapiro–Wilk test of normality but failed Levene’s test of homogeneity of variances; thus, ultimately, the Welch one-way ANOVA test (which does not require the assumption of homogeneity of variances) was used. All *p*-values were adjusted using the Benjamini–Hochberg (FDR) method. Plots were generated using the R package microeco [23]. For principal coordinates analysis (PCoA), confidence ellipses were generated with default parameters by the micreco package (trans_beta, plot_ordination), which uses the ggplot2::stat_ellipse method (default parameters: type = multivariate *t*-distribution, level = ellipse).

The association between demographic, environmental, and management factors and ESBL-E carriage was analyzed using univariable generalized estimating equation (GEE) with a logit link function; the heifers were defined as subjects, and the farm was defined as a within-subject effect. As ESBL-E carriers were sub-selected for this analysis, *n* = 150. For continuous variables (morbidity and mortality percentages), Spearman’s rho correlation was used. The SPSS statistical software (Version 24) was used for Spearman’s rho correlation and GEE analysis, and *p*-values < 0.05 were considered significant.

## 3. Results

### 3.1. General Characterization

Fecal samples (*n* = 157) were analyzed from eight dairy farms located within a 50 km radius in central Israel (Figure 1). All farms used a similar housing type (courts), with similar animal density (under five heads to every square meter). All farms used the same cleaning management practice for the animals and the environment. The reported prevalence of mastitis ranged between 10% and 30% in different farms (median = 18%, inter quartile range (IQR) = 13.6%). Reported calve morbidity ranged between 1% and 8.6% (median = 5%, IQR = 4%) and reported calf mortality ranged between 1% and 10% (median = 2%, IQR = 4.8%). No arrival of new cattle in the preceding month was reported by any of the farms. The nutritional composition of feed was similar in all farms; however, different ingredients were used by the farms (i.e., different types of cereals, grains, and pulps). All heifers were reported to be healthy during sampling by farm personnel.

Fifty-five heifers (35%) were ESBL-PE carriers and 102 (65%) were noncarriers. Of the 55 ESBL-PE-positive samples, 48 harbored ESBL-E (30.6% of all samples). A comparison between ESBL-PE carriers and noncarriers revealed significant differences between suckling of pooled colostrum and individual colostrum of the mother (*p* < 0.001), and between milk sources used for feeding calves (*p* < 0.001) (Table 1). ESBL-E carriage was also significantly associated with the use of antimicrobials (*p* = 0.003), with the presence of dogs on the premises (*p* = 0.02), and with the use of fans for cooling (*p* = 0.01) (Table 1).

The rates of morbidity (Rho = −0.329, *p* = 0.001) and mastitis (Rho = −0.364, *p* = 0.002) on the farms were weakly negatively correlated with ESBL-PE carriage. Mortality (Rho = −0.004, *p* = 0.799) was not significantly associated with ESBL-PE carriage. Prevalence of ESBL-PE carriers ranged between 0% and 100%, median 29% in all farms. In one farm, all sampled heifers were ESBL-PE-negative, and, in another farm, all sampled heifers were positive for ESBL-PE. More information on ESBL prevalence is provided in Appendix A. 

### 3.2. Alpha and Beta Diversity Analysis

Analysis of alpha diversity (Shannon diversity index) and beta diversity (unweighted UniFrac) between ESBL-PE carrier and noncarriers showed no significant difference (Figure 2). When comparing farms, we found a significant difference in both alpha and beta diversity (*p* < 0.001 and *p* = 0.001, respectively) (Figure 3A,B). When viewed via PCoA plot (Figure 3C), it was evident that some farms clustered together, while other farms (notably GI and V) clustered independently. These relationships did not appear to be related to the ESBL-PE carrier status of individual heifers. Post hoc testing also revealed significant differences between some but not all farms with regard to alpha and beta diversity. In addition, we found a significant difference in alpha diversity (Faith’s phylogenetic diversity (PD) measure) between farms feeding with pooled colostrum and individual mother colostrum (*p* = 0.037; mean = 31.65 and mean = 35.07, respectively), with a higher diversity of microbiota observed in individual mother colostrum (Figure 4). In total, 4669 taxa features were assigned (further information in Appendix A). 

### 3.3. Taxonomic Composition of Fecal Microbiota between ESBL-PE Carriers and Noncarriers

Microbiota analysis revealed that the most common bacterial phyla were Bacteroidetes (44%) and Firmicutes (38%). Inclusive of Archaea (Euryarchaeota phylum), the bacterial population comprised 20 phyla, 30 classes, 119 families, 270 genera, and over 600 species, of which over 50% were unidentified. The 10 most commonly identified phyla and their mean relative abundance by ESBL carrier status are listed in Table 2. There was no significant difference in taxonomic composition between ESBL-PE carriers and noncarriers at the phylum and genus level. However, using the LEfSe biomarker discovery method, we identified several genera which discriminated between ESBL-PE carriers and noncarriers, among the highest significantly different genera, with an LDA score >3. Prevotellaceae UCG-003, *Bacteroides*, and Rikenellaceae RC9 gut group were more abundant in the carrier group than the noncarrier group, while *Lysinibacillus*, Ruminococcaceae UCG-010, and *Treponema* 2 were more abundant in the noncarrier group than the carrier group. A detailed list is shown in Figure 5. Members of the Enterobacteriace family constituted only 0.14% of the bacterial population in both groups. 

## 4. Discussion

The key observation resulting from our study was a significant difference in certain genera which discriminated between ESBL-PE carriers and noncarriers. In the single study in cattle, in which the microbiome of 24 calves that were colonized with CRB was compared to the microbiome of 24 calves that were not colonized with CRB, differences were also observed in the abundance of various phyla, and it was suggested that the lower prevalence of CRB at age 9–12 months may have been associated with increased microbiota diversity in the gastrointestinal tract. However, a lower prevalence that was found in that study may have been the result of environmental factors such as temperature as was suggested by the authors [8]. The aim of this study was to compare the microbiome composition between ESBL-PE carriers and noncarriers, although this study cannot indicate the direction of the relationship due to its cross-sectional nature. 

Previous research observed differences in the cattle fecal microbiome according to age, breed, sex, geography, and management type [10]; as a result, in this study, we focused on a relatively homogenous group of animals in these aspects. Furthermore, we focused on an age group in which the colonization rate would be high enough to enable to select enough ESBL-PE-colonized and noncolonized animals. In previous studies, the colonization rate in calves in their first year of life was high [5,8,24] and gradually declined (by 6.5-fold) in adult cows. Therefore, we decided to test 12–18 month old heifers, and 35% were indeed ESBL-PE-positive, with 87.2% being carriers of ESBL-producing *E. coli*. This enabled us to select fecal samples from both ESBL-PE-colonized and no-colonized animals in each of the farms.

The results of this study further demonstrate the complexity of the colonization phenomenon. Although significant differences were not found between the microbiome composition of ESBL-PE carrier and noncarrier dairy heifers in this study, we were able to demonstrate that some aspects of microbiota composition did differ between the two groups, specifically the abundance of certain genera. Using LEfSe biomarker discovery analysis, we identified several genera which were significantly different between ESBL-PE carriers and noncarriers. *Prevotella* spp. and Bacteroidales spp. were more abundant in the ESBL-PE carrier group and, therefore, could be considered as a potential marker for ESBL-PE carriers, as was suggested previously [12,16]. *Lysinibacillus*, Ruminococcaceae and *Treponema* were more abundant in the noncarrier group and may also serve as potential markers. Further research is needed to assess these potential markers and their reproducibility. 

Another observation in this study was a larger number of different taxonomy levels in the cattle fecal microbiome than previously reported [11], which may have resulted from differences in methodology, as well as differences in geography and nutrition. According to a meta-analysis of bacterial diversity in cattle, Firmicutes is the most represented phylum, followed by *Bacteroides* and *Proteobacteria* [11]. This is in line with our study, in which Bacteroidetes and Firmicutes accounted for 84% of the microbiota. This is also supported by several other studies in cattle, other animals, and humans [11,12,15,24,25,26]. In our study, *Bacteroidia* was the largest class with the most genera featured, while Ruminococcaceae and Rikenellaceae were the largest families, which are different findings than previously reported [11,25]. 

We found very low abundance of the Enterobacteriaceae family, with the genus *E. coli* constituting only a very small fraction of total sequences. This is in accordance with previous findings in French Guinea [14] and France [15]. This low abundance raises some questions regarding the influence of this bacterial family on the microbiome and whether ESBL-PE have sufficient metabolic activity to influence microbial population or vice versa. Furthermore, a study on ARGs in the feces of dairy cattle showed that Enterobacteriaceae constitute approximately 25% of the microbiota in the first week of life; however, soon afterward, the relative abundance decreased significantly to less than 5% [27]. Furthermore, a longitudinal cohort investigating the microbial assembly of rumen cattle from birth to adulthood identified changes in the microbiome which were directly linked to the animal age [28]. They also found that the assembly of the microbiome during the first 24 h of life affected the dynamics and composition of the microbiome throughout the life of the individual animal [28]. This links to another finding in this study in which we observed that the microbiome composition was significantly different between calves that were fed with pooled colostrum compared to individual mother colostrum, regardless of the colonization status. Since this difference was found to be significant between farms and individual farms only fed with pooled colostrum or individual colostrum, these results should be interpreted with caution.

This study had several limitations. First, we used multiple farms and found a significant difference in alpha and beta diversity between them; however, due to sample size, we did not adjust analysis by farm. Although the farms were supposedly similar in most aspects of housing and management (we only found a difference in colostrum intake and feeding method for calves), it would have been better if all animals were from the same farm. Second, we did not exclude farms with antibiotic exposure; however, only a few heifers received a single dose of antibiotics, and the sampling point did not coincide with the antibiotic intake allowing microbiome composition to return to normal. Additionally, statistical analysis showed no significant difference, suggesting that this exposure was of minimal impact on colonization rates. Third, this study only provided information on the characteristics of the fecal microbiome with known colonization, but explained neither the initial changes that occur nor the mechanisms leading to the acquisition of these resistant strains, as a longitudinal design could have provided. Future studies should consider a longitudinal approach allowing the analysis of microbiota dynamics over time on larger farms or a greater number of farms, as well as establishing the directionality of the relationships between ESBL-PE status and the fecal microbiome. Furthermore, future studies should delve more deeply into the effect of contaminants in feed that could potentially drive AMR, as well as compositional elements of feed which may alter the overall fecal microbiome [29,30,31]. 

## 5. Conclusions 

The results of this study further demonstrate the complexity of the colonization phenomena associated with antimicrobial-resistant bacteria among dairy farms. When conducting beta diversity analysis, it was observed that the microbiome of ESBL carriers exhibited lower diversity than the microbiome of noncarriers. Furthermore, we were able to demonstrate that some aspects of microbiota composition do differ between the two groups, specifically the abundance of certain genera (i.e., *Prevotellacaea, Bacteroides*, Rikenellaceae, and uncultured Bacteroidales*)*. These could potentially serve as markers of ESBL-PE carriage. However, it is questionable whether these differences have a significant impact on colonization with ESBL-PE. Further research is needed to establish the utility of these potential markers, as well as to uncover further relationships between ESBL-PE carriage and the microbiome. 

## Figures and Tables

**Figure 1 animals-12-01738-f001:**
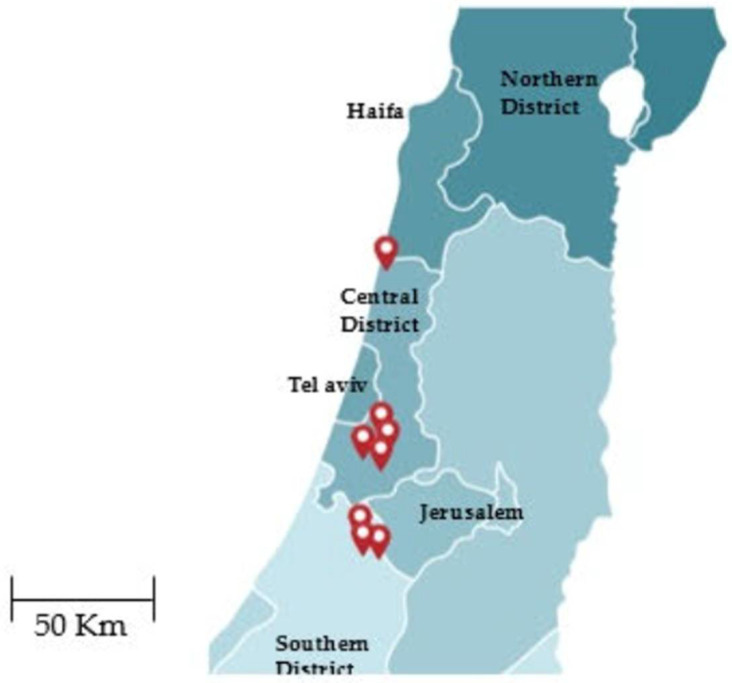
Location of eight sampled dairy farms in central Israel.

**Figure 2 animals-12-01738-f002:**
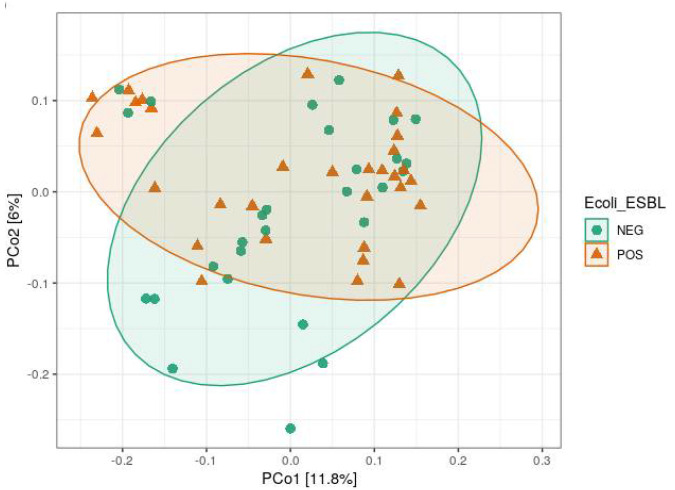
Beta diversity (unweighted UniFrac) comparison of ESBL-PE noncarriers (NEG) with ESBL-PE carriers (POS) via principal coordinates analysis, with confidence ellipses.

**Figure 3 animals-12-01738-f003:**
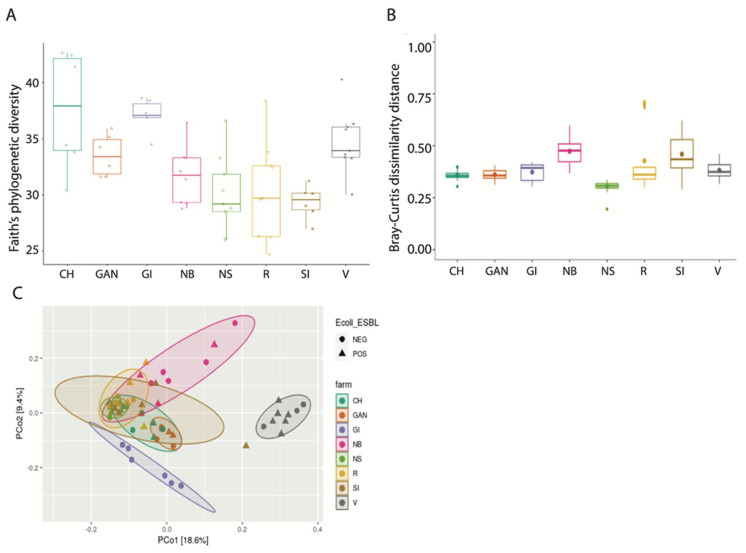
Alpha and beta diversity among farms. Symbols in figures represent individual data points. (**A**) Alpha diversity among farms (ANOVA test with Benjamin–Hochberg (FDR) method, overall *p* = 0.001). (**B**) Beta diversity Bray–Curtis comparison among farms (PERMANOVA (using 999 permutations), overall *p* = 0.001). (**C**) Principal coordinates analysis (PCoA) of beta diversity taking into account *E. coli* ESBL carrier status, with confidence ellipses.

**Figure 4 animals-12-01738-f004:**
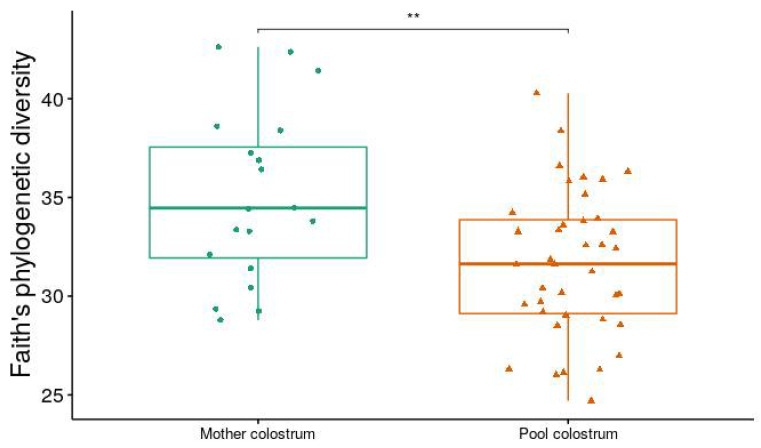
Faith’s phylogenetic diversity (PD) between microbiomes of heifers fed individual mother colostrum and those fed pooled colostrum (Wilcoxon test, with Benjamin–Hochberg (FDR) method, *p* = 0.037). Symbols in figures represent individual data points. Beta diversity between these two groups was not significant. ** *p* < 0.01.

**Figure 5 animals-12-01738-f005:**
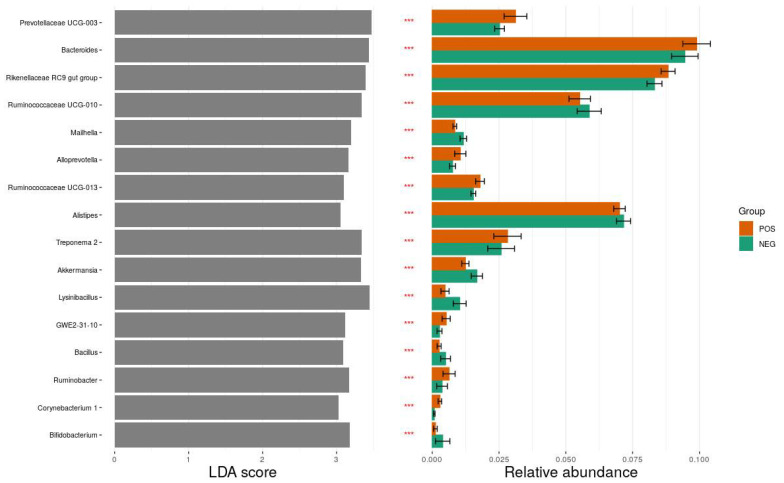
Differential abundance analysis between *E. coli* ESBL-positive (colored orange) and *E. coli* ESBL-negative (colored green) groups using the linear discriminant analysis (LDA) effect size (LefSe) method. Sixteen significant taxa are presented, with LDA scores >3 (left plot), statistical significance < 0.05 (asterisks in the middle column), and relative abundance (right plot). *** *p* < 0.001.

**Table 1 animals-12-01738-t001:** Analysis of the association between demographic, environmental, and management factors, and extended-spectrum β-lactamase-producing *Enterobacterales* (ESBL-PE) carriage in dairy heifers. The analysis was performed using a univariable GEE model with the farm defined as a random factor.

		*N*	*N* Positive (%)	OR (95% CI)	*p*-Value
Cooling system	Fan	37	16 (43.2%)	0.59 (0.39–0.88)	0.01
	Nebulizers	20	2 (10%)	1.29 (0.60–2.79)	0.506
	Both	100	37 (37%)	ref	ref
Working dogs (yes/no)	No	20	2 (10%)	ref	ref
	Yes	137	53 (38.7%)	5.68 (1.27–25.47)	0.023
Colostrum feeding of calves	Pooled colostrum	97	48 (49.5%)	7.42 (3.07–17.94)	<0.001
	Individual colostrum	60	7 (11.7%)	ref	ref
Milk feeding of calves	Milk replacement	80	17 (21.3%)	0.096 (0.04–0.24)	<0.001
	Whole milk	39	10 (25.6%)	0.12 (0.04–0.34)	<0.001
	Both	38	28 (73.7%)	ref	ref
Method for manure cleaning	Tractor	79	32 (40.5%)	0.85 (0.30–2.39)	0.76
	Automatic shovel	60	15 (25%)	0.42 (0.14–1.25)	0.118
	Both	18	8 (44.4%)	ref	ref
Antimicrobial prophylaxis	No	117	33 (28.2%)	ref	ref
	Yes	40	22 (55%)	3.11 (1.48–6.53)	0.003

**Table 2 animals-12-01738-t002:** Summary of 10 most common phyla in ESBL-PE carriers and noncarriers.

	ESBL-PE-Negative			ESBL-PE-Positive			
Phylum	*N*	Mean	SD	*N*	Mean	SD	*p*-Value
Bacteroidetes	26	0.439	0.045	33	0.458	0.051	0.621
Firmicutes	26	0.397	0.04	33	0.386	0.045	0.801
Spirochaetes	26	0.029	0.03	33	0.034	0.036	0.986
Proteobacteria	26	0.03	0.013	33	0.029	0.017	0.801
Tenericutes	26	0.021	0.008	33	0.021	0.008	0.977
Verrucomicrobia	26	0.022	0.007	33	0.015	0.007	0.801
Cyanobacteria	26	0.017	0.011	33	0.013	0.007	0.621
Kiritimatiellaeota	26	0.014	0.006	33	0.011	0.006	0.977
Actinobacteria	26	0.012	0.01	33	0.005	0.009	0.977
Lentisphaerae	26	0.005	0.01	33	0.004	0.008	0.801

*N* = number of samples, Mean = relative abundance, SD = standard deviation.

## Data Availability

Data are available in the NCBI BioProject Database, Bioproject ID: PRJEB54070.

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
