# Peer review of "Fecal Microbiome Features Associated with Extended-Spectrum β-Lactamase-Producing Enterobacterales Carriage in Dairy Heifers"

_animals, 2022, doi:10.3390/ani12141738_

Round 1
Reviewer 1 Report
In my point of view, this article is not improved essentially. First of all, proper methods, as well as informative descriptions of it, are fundamental to draw objective results. There are still some fatal problems regarding data analysis. Here are some typical examples.
1) Figure 2A, Figure4, Table 2, Figure 5
Since we can see obvious difference among farms in Figure 3C, farm effect should be considered or adjusted when testing microbiome differences (including alpha diversity, beta diversity, and differential taxa) between ESBL-PE carriers and non-carriers.
2) Figure 2A, Figure 3B
Permutational multivariate analysis of variance (PERMANOVA) or analysis of similarities (ANOSIM) should be used when testing the significance of difference in beta diversity, rather than ANOVA or Wilcoxon test.
3) Table 1
How was independent variable (0/1, variable indicating ESBL-PE carriers or non-carriers) defined in the GEE model? The model of GEE should be fully described or using a mathematical equation.
4) Table 2, Figure 5
Is there any difference in the objective of the two analyses? I thought they were duplicated, and a more suitable one should be used and further revised.
# Line 257-260: There was no significant difference in taxonomic 258 composition between ESBL-PE carriers and non-carriers at the phyla and genera level. However, using the LEfSe 259 biomarker discovery method, we identified several genera which discriminated between ESBL-PE carriers and non-260 carriers, among the highest significantly different genera, with an LDA score greater than 3.
What was the threshold of p-value or adjusted p-value in LEfSe analysis? This was not mentioned in the methods. If you used the same p-value threshold for Wilcoxon and LEfSe analysis, these should be some consistent results between two methods. Still, how was the farm effect considered when conducting differential abundance analysis?
In conclusion, I thought this article did not meet the criteria for publication give the substantial concerns with methods and results.
Author Response
1) Figure 2A, Figure4, Table 2, Figure 5
Since we can see obvious difference among farms in Figure 3C, farm effect should be considered or adjusted when testing microbiome differences (including alpha diversity, beta diversity, and differential taxa) between ESBL-PE carriers and non-carriers.
Due to the sample size, we did not adjust by farm in the analysis. We have amended the limitations accordingly.
2) Figure 2A, Figure 3B
Permutational multivariate analysis of variance (PERMANOVA) or analysis of similarities (ANOSIM) should be used when testing the significance of difference in beta diversity, rather than ANOVA or Wilcoxon test.
We have amended with PERMANOVA analysis of beta diversity and adjusted our presentation and description of results accordingly.
3) Table 1
How was independent variable (0/1, variable indicating ESBL-PE carriers or non-carriers) defined in the GEE model? The model of GEE should be fully described or using a mathematical equation.
As mentioned in our previous revision, we used univariate GEE analysis with binary ESBL-carrier status as the outcome. Multivariate GEE was removed in the last revision. The univariate parameters are defined accordingly in the methods.
4) Table 2, Figure 5
Is there any difference in the objective of the two analyses? I thought they were duplicated, and a more suitable one should be used and further revised.
# Line 257-260: There was no significant difference in taxonomic 258 composition between ESBL-PE carriers and non-carriers at the phyla and genera level. However, using the LEfSe 259 biomarker discovery method, we identified several genera which discriminated between ESBL-PE carriers and non-260 carriers, among the highest significantly different genera, with an LDA score greater than 3.
What was the threshold of p-value or adjusted p-value in LEfSe analysis? This was not mentioned in the methods. If you used the same p-value threshold for Wilcoxon and LEfSe analysis, these should be some consistent results between two methods. Still, how was the farm effect considered when conducting differential abundance analysis?
The same FDR p-value (0.05) threshold was used, as stated in the methods. LEfSe is a microbiome-oriented analysis noted to perform better with microbiome data than standard statistical tests, so it is not surprising that the results differed between standard statistical tests and LEfSe analysis (Segata et al.: Metagenomic biomarker discovery and explanation. Genome Biology 2011 12:R60.)
Reviewer 2 Report
The authors have revised the manuscript effectively. Also, they have made modifications in the manuscript as suggested previously. I believe that after the corrections suggested by me and other reviewers the article may be accepted for publication.
Author Response
We thank the reviewer for their comments and suggestions!
Reviewer 3 Report
Cohen et al comments v2
This version of the manuscript is improved and the additions to the methods are sufficient. The figures and tables are low quality and should be improved before publication. Figures are generally low resolution and look to be created in R. High resolution figures can easily be exported by running tiff("FILENAME.tiff", units="in", width=5, height=5, res=400) before your code and then dev.off() directly after your graphic code. Figure 3 does not have all panels A, B, and C labeled in a consistent manner easily recognizable to the reader. Each panel within Figure 3 was generated using a different ggplot theme and font sizes, they should at the very least be consistent within a single Figure to make it easier for the reader to follow. Table 2 is in a completely different format than Table 1. Figure 5 should have both panels labeled A and B and this reflected in the caption.
L92 – “stocking density” is the appropriate technical term to replace crowdedness
L125 – do you mean 806R? I am not aware of 807R primers
L129 – was this data processing done using Illumina’s built-in platform or on QIIME2? You repeat this information later L137-141 so I am unsure of the steps here
L224 – Principal not principle, what are the ellipses? Are they confidence ellipses and if so how were they calculated?
L242 – Principal, explain ellipses
L269 – is this part of the caption?
Author Response
This version of the manuscript is improved and the additions to the methods are sufficient. The figures and tables are low quality and should be improved before publication. Figures are generally low resolution and look to be created in R. High resolution figures can easily be exported by running tiff("FILENAME.tiff", units="in", width=5, height=5, res=400) before your code and then dev.off() directly after your graphic code. Figure 3 does not have all panels A, B, and C labeled in a consistent manner easily recognizable to the reader. Each panel within Figure 3 was generated using a different ggplot theme and font sizes, they should at the very least be consistent within a single Figure to make it easier for the reader to follow. Table 2 is in a completely different format than Table 1. Figure 5 should have both panels labeled A and B and this reflected in the caption.
We have edited to improve figure and table quality and consistency.
L92 – “stocking density” is the appropriate technical term to replace crowdedness
We have amended accordingly
L125 – do you mean 806R? I am not aware of 807R primers
We have amended accordingly.
L129 – was this data processing done using Illumina’s built-in platform or on QIIME2? You repeat this information later L137-141 so I am unsure of the steps here
We have amended the methods section accordingly.
L224 – Principal not principle, what are the ellipses? Are they confidence ellipses and if so how were they calculated?
L242 – Principal, explain ellipses
We have amended the typo accordingly and included “with confidence ellipses” in the figure legend and explained ellipses calculation in the methods.
L269 – is this part of the caption?
Yes, we have amended accordingly to clarify.